# Two-Dimensional Ti_3_C_2_ MXene-Based Novel Nanocomposites for Breath Sensors for Early Detection of Diabetes Mellitus

**DOI:** 10.3390/bios12050332

**Published:** 2022-05-13

**Authors:** Anna Rudie, Anna Marie Schornack, Qiang Wu, Qifeng Zhang, Danling Wang

**Affiliations:** 1Department of Electrical and Computer Engineering, North Dakota State University, Fargo, ND 58102, USA; anna.rudie@ndsu.edu (A.R.); qifeng.zhang@ndsu.edu (Q.Z.); 2Department of Physiology and Biomedical Engineering, Mayo Clinic, Rochester, MN 55902, USA; schornack.annamarie@mayo.edu; 3Department of Mathematics, Physics, and Electrical Engineering, Northumbria University, Newcastle NE1 8ST, UK; qiang.wu@northumbria.ac.uk; 4Materials and Nanotechnology Program, North Dakota State University, Fargo, ND 58102, USA

**Keywords:** 2D Ti_3_C_2_ MXene, 1D nanostructured semiconductors, breath acetone, diabetes, chemiresistive sensor

## Abstract

The rates of diabetes throughout the world are rising rapidly, impacting nearly every country. New research is focused on better ways to monitor and treat this disease. Breath acetone levels have been defined as a biomarker for diabetes. The development of a method to monitor and diagnose diabetes utilizing breath acetone levels would provide a fast, easy, and non-invasive treatment option. An ideal material for point-of-care diabetes management would need to have a high response to acetone, high acetone selectivity, low interference from humidity, and be able to operate at room temperature. Chemiresistive gas sensors are a promising method for sensing breath acetone due to their simple fabrication and easy operation. Certain semiconductor materials in chemiresistive sensors can react to acetone in the air and produce changes in resistance that can be correlated with acetone levels. While these materials have been developed and show strong responses to acetone with good selectivity, most of them must operate at high temperatures (compared to RT), causing high power consumption, unstable device operation, and complex device design. In this paper, we systematically studied a series of 2-dimensional MXene-based nanocomposites as the sensing materials in chemiresistive sensors to detect 2.86 ppm of acetone at room temperature. Most of them showed great sensitivity and selectivity for acetone. In particular, the 1D/2D CrWO/Ti_3_C_2_ nanocomposite showed the best sensing response to acetone: nine times higher sensitivity than 1D KWO nanowires. To determine the sensing selectivity, a CrWO/Ti_3_C_2_ nanocomposite-based sensor was exposed to various common vapors in human breath. The result revealed that it has excellent selectivity for acetone, and far lower responses to other vapors. All these preliminary results indicate that this material is a promising candidate for the creation of a point-of-care diabetes management device.

## 1. Introduction

The diabetes epidemic is growing throughout the world and has large impacts on every country. In 2020, there were 463 million people throughout the world living with diabetes. Predictions suggest 578 million people will have diabetes by 2030. Diabetes, if not well managed, can lead to serious health complications and is among the top 10 causes of death globally [1]. In 2019, diabetes was directly related to 1.5 million deaths, making it the ninth leading cause of death [2]. The rise of the diabetes epidemic suggests there is an immediate need for better, more accessible ways of monitoring, diagnosing, and treating this disease even at early stage. To maintain good control over the disease, patients with diabetes need to monitor their glucose levels closely. It is recommended that blood-glucose levels be tested 4 to 10 times a day with type I diabetes [3]. Currently, the most prevalent way of monitoring glucose levels in patients with diabetes is through blood-glucose monitoring using portable glucose meters. A sensing strip is inserted into a glucometer and a drop of blood from the patient’s finger is placed on the strip. This method, while accurate, is painful, invasive, and carries a high risk of infection [4]. In addition, the strips used for testing the blood-glucose levels are not reusable. They are expensive to replace over time. While continuous blood-glucose monitoring is becoming more popular as the devices’ accuracy increases and as they become more convenient to use, this method can cause unavoidable issues, as it is expensive, invasive, requires a device that is a little bit bulky and has a limited lifetime, and is uncomfortable [5]. Beside these drawbacks, some continuous blood-glucose monitoring devices require calibration using capillary blood-glucose meters, which is inconvenient and more expensive. Additionally, false alarms of hypo- or hyperglycemia can be incredibly inconvenient to the users [6]. Urinalysis is another way to monitor blood sugar levels, though it is inconvenient, embarrassing, and cannot be done everywhere. Breath-based biosensors have the advantages of being non-invasive, easily repeated, and neither painful nor embarrassing like the blood and urine tests [7].

Human breath is complicated and composed of nitrogen, oxygen, carbon dioxide, water, inert gases, and trace volatile organic compounds (VOCs). Some VOCs can be linked to metabolic processes in the body [8]. Acetone is one such VOC that has been correlated with blood-glucose levels in the body. People with a high level of glycemia have acetone content exceeding 0.9 ppm on their breath [9]. A previous study showed that a breath acetone concentration of 1 ppm was only seen in cases of serious stages of diabetes [10]. Other studies found a concentration of 0.8 ppm in control groups and 1.8 ppm for those with diabetes [4,11]. The most popular ways of detecting trace chemicals on human breath involve gas chromatography with flame ionization detection, mass spectrometry, and ion mobility spectrometry [4,8,12]. The problem with these methods is the size and cost of the equipment needed to run the breath tests. In addition, these tests cannot be performed anywhere other than a laboratory.

For clinical applications, a high-performance breath sensor should have high sensitivity, excellent selectivity, a fast response, and immunity from the interference of water vapor, since the relative humidity of human breath is 80% or higher [12]. While complete selectivity of a sensor to only one gas goes against the nature of semiconductor-based sensors, an ideal sensor for a breath analyzer would show a much higher response to the target gas than to other gases [13,14,15]. Additionally, the device itself should be hand-held, easy to operate, and have low construction and maintenance costs [13]. With the developments in nanotechnology and nanomaterial synthesis, nanostructured semiconductor-based gas sensors have been widely developed and provided promising applications in breath analysis due to their simple fabrication, high sensitivity, easy operation, miniaturization, and great integration. This type of sensor follows the working mechanism of chemiresistive responses. That is, a change in resistance of the sensor can be measured by an ohmmeter when the sensor is introduced to a specific gas molecule [15]. The nanostructured semiconductors offer tunable structures and can introduce new material properties to significantly improve the selectivity issue compared to traditional semiconductor-based chemiresistors. However, the major problem in most nanostructured semiconductor-based sensors is the operating temperature, which constitutes a critical technical drawback [4] and limits their application in breath analysis. This is because (1) the increase in resistance at a high temperature causes lower sensitivity and stability of the sensing material; (2) a high operating temperature requires the integration of a micro-heater into the sensor device and makes for a complicated detecting system [16]. To tolerate high temperatures, both the micro-heater and the electrodes for the resistance measurement are made with Au or Pt. Together with the complicated configuration of the devices required to allow for heating, this makes them expensive both in terms of materials and manufacturing; (3) a high operating temperature makes it difficult to integrate the sensor component into a circuit; and (4) a high operating temperature increases power consumption. Additionally, semiconductor nanomaterial-based gas sensors have poor tolerance of humidity. The high humidity in exhaled breath can interfere with and degrade their sensing responses [17]. All these challenges indicate the limitations of traditional and nanostructured semiconductor-based gas sensors for applications in point-of-care disease detecting and monitoring through breath analysis.

In our previous study [16,17,18,19,20,21,22,23], 1-dimensional (1D) nanowires, K_2_W_7_O_22_ (KWO), were successfully synthesized and showed unique room-temperature ferroelectric properties which caused strong attraction towards polar gases, such as acetone, due to a strong charge transfer between acetone and ferroelectric KWO. The KWO nanomaterial also has a large surface to volume ratio due to the nanowire structure [22]. The successful synthesis of KWO motivated us to study the origin of room-temperature ferroelectricity in KWO and find further improvements for application in breath acetone detection. In addition, like other nanomaterials, the structure of KWO is tunable, which means that the lattice structure of the material can be adjusted and reconfigured based on the interactions with electromagnetic composites.

Our group successfully synthesized a new 2-dimentional (2D) nanomaterial, Ti_3_C_2_T_x_ (T_x_ stands for OH^−^, –O, and F^−1^ surface terminated groups) MXene [20,23,24], named from the general formula of M_n+1_X_n_ (*n* = 2, M = Ti, and X = C) [25,26]. Due to its unique multi-layered structure with an extremely large surface and interface area, metallic conductivity, high signal to noise ratio, and flexible surface functionality, 2D MXene Ti_3_C_2_T_x_ has attracted a lot of attention in the fields of biomedical sensing and energy storage [25,26,27,28,29,30,31,32]. Our preliminary results indeed revealed the 1D/2D KWO/Ti_3_C_2_ nanocomposites exhibit much better sensing performance for detecting acetone than KWO nanowires in terms of acetone detection and humidity tolerance [19,20,21,22]. Motivated by this, in this paper, we introduce a series of new nanocomposites composed of one-dimensional (1D) MWO nanowires and two-dimensional (2D) Ti_3_C_2_ MXene nanosheets: the sensing materials to detect acetone at room temperature. Here, MWO refers to 1D semiconducting nanowires, M_x_W_7_O_22_ (x = 2), and M includes Li, Na, K, Cr, and a combination of K/Li. These metals were chosen because Li, Na, and K all belong to group 1 on the periodic table, meaning they have similar electrical properties but different atomic sizes. Their sensitivity, stability, and selectivity in MWO and nanocomposites—partly to understand the origin of their ferroelectricity—for acetone sensing, were studied in order to develop the ideal breath acetone sensor for the application of point-of-care diabetes monitoring and early diagnosing.

## 2. Materials and Methods

The 1D KWO nanowires were synthesized utilizing a hydrothermal method. A precursor solution is made from di-H_2_O, oxalic acid dehydrate (>99%, VWR, Radnor, PA, USA), K_2_SO_4_ (>99%, VWR, Radnor, PA, USA), Na_2_WO_4_·2H_2_O (95%, Alfa Aesar, Tewksbury, MA, USA), and hydrochloric acid (36–38%, Aqua Solutions Inc., Deer Park, TX, USA). The solution is heated in a 30 mL autoclave at 225 °C for 24 h. The resulting product is KWO nanowires [18,19,20,21]. 

The MWO nanomaterial was synthesized similarly to KWO. Li_2_SO_4,_ Na_2_SO_4_, CuSO_4_, and Cr_2_(SO_4_)_3_, or combinations of the metal sulfates, were used in place of K_2_SO_4_ to result in the product of M_x_W_7_O_22_. The remainder of the protocol remained the same as the previously mentioned one for the synthesis of KWO. The various concentrations of metal sulfates were mixed with di-H_2_O, oxalic acid dehydrate (>99%, VWR, Radnor, PA, USA), Na_2_WO_4_·2H_2_O (95%, Alfa Aesar, Tewksbury, MA, USA), and hydrochloric acid (36–38%, Aqua Solutions Inc., Deer Park, TX, USA) to create a precursor solution. The solution was then heated to 225 °C for 24 h in a 30 mL autoclave. 

The 2D Ti_3_C_2_ nanosheets were prepared using the “hot etching method” developed in the laboratory [32]. The Ti_3_AlC_2_ MAX phase was obtained through ball milling Tic, Ti, and Al powders for 5 h using the molar ratio 2:1:1.2, respectively. The resulting powder was pressed into a pellet under argon flow and sintered at 1350 °C for 4 h. The pellet was then milled back into powder form and sieved through a 160-mesh sieve. The MAX powder was collected at elevated temperatures for etching. Hydrofluoric (HF) acid was heated in a 25 mL Teflon line autoclave at 150 °C for 5 h to etch the MAX phase. Then, 0.5%wt of HF was used to remove Al from the MAX phase. The materials are then sonicated for one hour and collected through centrifuge. The material was left to dry overnight in a drying oven at 65 °C.

The synthesis of 1D/2D MWO/Ti_3_C_2_ nanocomposites followed the steps: (1) a certain amount of as-synthesized KWO nanorods were dispersed into distilled water and sonicated for 4 h. Then, (2) as-synthesized Ti_3_C_2_ nanosheets were added to the distilled water treated by sonication. Thereafter, the homogeneous MWO solution was poured into the Ti_3_C_2_ solution and stirred vigorously to obtain well mixed MWO/Ti_3_C_2_ powder after post washing and drying. 

The sensor slides were generated by coating a gold-plated slide with the nanomaterial. The slide was approximately 1″ × 1″, and the gold plating created a ladder design to ensure that the entire area of the nanomaterial received the same amount of voltage. The sensing area was approximately 0.8″ × 0.66″, and electrode diameter was 0.125″. The gap between the interdigitated electrodes was 0.25″.

To conduct the tests, each slide was placed into a sealed testing box. This box had pogo pin connectors that made contact with pads on the slides to measure resistance. Controlled concentrations of gas were pumped into the testing box using an OVG-4 vapor and OHG-4 humidity generator. For these tests, the concentration of acetone was 2.86 ppm. For the selectivity testing, ethanol, methanol, toluene, and water were tested at the same concentration of 2.86 ppm. Changes in resistance across the sensor were measured using a Keithley electrometer. The procedure for testing involved an initial resistance measurement before the slides’ environment was manipulated. Dry air was pumped into the box until the RH of the box was lowered to between 20% and 30% RH. The goal of this was to create an environment with a low RH that would remain stable throughout the test. Once the RH was at the correct level, the resistance of the slide was measured, the dry air was turned off, and acetone was pumped into the container for a total of 2 min. After the 2-min time interval, the acetone was turned off, the resistance was measured again, and the box was opened to allow the slide to return to its baseline value.

As mentioned briefly above, a good candidate for a breath-acetone sensor should have a strong and fast response to acetone, and high acetone selectivity. Sensitivity is the first parameter we look at and can be determined by the equation below:(1)$$S\%=\left(\frac{R_{F}-R_{A}}{R_{I}}\right)*100,$$
where *R_F_* is the final resistance, measured after the sensor is exposed to acetone for two minutes, *R_A_* is the resistance after the relative humidity (RH) inside the testing box is lowered to the appropriate value, and *R_I_* is the initial resistance, measured at room temperature and room RH. Previous research has found that KWO-based nanosensors were more responsive to acetone when operating at lower RH. To avoid this factor, in our experiments, the testing process initially focused on eliminating the interference of RH to provide more accurate sensing results, and we created a new equation (Equation (1)) with which to calculate the sensitivity. This equation is different than other equations (Equations (2) and (3)) used in the literature [16,19] for sensitivity calculations for p-type and n-type metal oxide-based sensors. For p-type metal oxide-based sensors:(2)$$R=\frac{R_{gas}}{R_{air}}.$$

For n-type metal oxide-based sensors:(3)$$R=\frac{R_{air}}{R_{gas}},$$
This is because these equations do not consider the initial resistance of the material, so the calculations can produce vastly different results. Instead, ours (Equation (1)) is a more accurate calculation of the actual sensitivity of the material because the initial resistance is important when considering the strength of the signal received when gas is introduced into an ambitious environment with variable RH, specific to human breath.

## 3. Results and Discussion

### 3.1. Electron Characterization of Sensing Materials

Most metal oxide-based sensors rely on a redox reaction, and therefore need high operating temperatures for gas detection. KWO is a unique sensing material that utilizes its room-temperature ferroelectric property for acetone detection without assistance from an elevated temperature. In our previous study, we also identified that KWO is a p-type semiconductor, meaning the majority of its carriers are holes. Acetone is polar and an electron-rich compound, which makes it very attracted to the charged crystal faces of p-type KWO. Once they contact each other, an efficient electron transfer is immediately induced and quickly distributes into a porous KWO sensing film. Electrons and holes formed electron–hole pairs and hole concentration in KWO decreased, which eventually caused an increase in the resistance of KWO [16,17,18,19,20,21,22,23]. Our previous work has also shown that the ferroelectric property of the material is dependent on the processing and synthesis of the material [20,21]. Therefore, we introduced other metals into the sensing material synthesis and formed MWO and MWO/Ti_3_C_2_ nanocomposites to better understand the structure–property relationships in their sensing responses. The SEM images of MWO and nanocomposites, as shown in Figure 1, all indicate similar nanowire morphologies, and the nanocomposite of NaWO/Ti_3_C_2_ (2:1) had a similar morphology to KWO/Ti_3_C_2_ (2:1), reported previously [20].

### 3.2. Sensitivity Testing

Of the sensors tested, CrWO: Ti_3_C_2_ (2:1) showed the highest sensitivity in response to acetone (Figure 2). This suggests that the involvement of chromium (Cr) in nanomaterial synthesis to form CrWO/MXene nanocomposites not only introduces more active sites, due to the hybrid structure, to interact with acetone molecules, but also could result in a stronger ferroelectric property which is a similar explanation to Cr-doped WO_3_ to detect acetone [33], but details are still under investigation. While the results point to this as a possible explanation, further testing and research need to be conducted.

The improvements in the sensing performance of various materials have been attributed to several things: band bending due to fermi level equilibration; charge carrier separation; depletion layer manipulation; increased interfacial potential barrier energy; chemical effects such as decrease in activation energy, targeted catalytic activity, and synergistic surface reactions; and geometrical effects such as grain refinement, surface area enhancement, and increased gas accessibility [34]. As such, sensing performance can be affected by the semiconducting type, microstructure, grain size, number of activated absorption sites, gas diffusion, effects of impurities, effect of heterojunction, and effects of humidity [35]. Previous tests of various materials produced results that are comparable to those achieved from this experiment (Table 1).

### 3.3. Semiconductor Resistance

There are three main functions that contribute to a p-type semiconductor’s resistance change: the receptor function, the transducer function, and the utility factor. The receptor function is concerned with the sensitivity and selectivity of a semiconductor and how each semiconductor interacts with the absorbed gas molecules. For the materials above, oxygen’s interaction with the surface of the material changes the electronic properties and allows for interaction with an analyte gas. This can be impacted by the specific elements of the material [41]. The transducer’s function has to do with the specific semiconductive properties of the material and how changes in the surface create changes in resistance of the material. Chemical interactions with the surface of the semiconductor create electric signals that depend on the surface potential and potential barriers formed between grains. The utility function is related to the morphological structure of the material and the diffusion and reaction of a target gas through the structure pores. Porous layers have a large gas-active surface area which allows for a larger reaction with a target gas [42,43]. These functions all explain why some materials perform better in sensing testing than others. 

Other research has shown the effect that alkali metals have on WO_3_. The bandgap of a material is the gap between the valence band of electrons and the conductive band. Larger gaps require more energy to move an electron from one band to the other to produce a response from the material. Materials with bandgaps of 4 eV or larger are considered insulators, though that cutoff number is somewhat arbitrary. Materials with nearly zero bandgaps are conductors. The materials with bandgaps above 0 eV but less than 4 eV are semiconductors, and the size of the band gap of such a semiconductor has some effect on its electric properties. According to Tosoni S. et.al.’s report [44], the band gaps of WO_3_ doped with various alkali metals are lower than the bandgap of WO_3_ without doping. These indicate different charge transfer capabilities which are correlated with the sensitivity of those materials. In our experimental results in Table 2, we prove the relationship between bandgap and sensitivity: a lower bandgap meant a higher sensitivity to 2.86 ppm acetone at the same sensing temperature. In addition, our results revealed that not only the size of the bandgap but also the different alkali metal molecules can influence the materials’ sensing properties, which is consistent with the results reported in other groups [45,46,47]. In Table 2, there are no big differences in the bandgaps in LiWO, NaWO, and KWO. However, LiWO had the highest sensing response to acetone. Further investigation is needed to understand the role of alkali metals in WO_3_.

### 3.4. Selectivity Testing

Due to the large number of VOCs in the breath, the selectivity of the material is crucial. Various sensors were tested to determine their responses to the presence of acetone, ethanol, methanol, toluene, and water at room temperature. All of tested gases were generated from OVG-4 and OHG-4 with the same controlled concentration, 2.86 ppm. The results show that CrWO/Ti_3_C_2_ is the material with the highest response to acetone, but also much better selectivity to acetone than the other chemicals tested (Figure 3). This means the 1D/2D nanocomposite CrWO/Ti_3_C_2_ is an even more promising material than KWO/Ti_3_C_2_ for the detection of acetone. Regarding the aforementioned theory about band gap, the hybrid structure caused ferroelectricity, and the receptor, transducer, and utility factors in this nanocomposite could be potential reasons for such a significant improvement in the sensing response to acetone, although further mechanistic details are still under investigation. 

## 4. Conclusions

Progress has been made to improve sensors’ performance to detect breath acetone using Ti_3_C_2_ MXene-based nanomposites. A 1D/2D nanocomposite, CrWO/Ti_3_C_2_, displayed highest sensitivity to 2.86 ppm acetone at room temperature; its detection limit was 0.1 ppm. Its sensitivity is almost nine times higher than that of the KWO/Ti_3_C_2_ nanocomposite to 2.86 ppm acetone at room temperature [20]. Such a high performance has not been shown in previous work. According to the sensing results, the proposed sensing mechanism could be explained by (1) stronger ferroelectricity created by the Cr-involved unsymmetrical structure of WO_3_ and the interfaces between 1-dimensional CrWO nanowires and 2-dimensional Ti_3_C_2_ nanosheets; (2) the excellent conductivity of nanocomposites; and (3) the huge surface and interface surface and rich active sites because of hybrid structures. There is still research to be conducted in order to understand the sensing mechanism and the relationships between sensing material structures and properties. These will guide sensing material synthesis and device optimization for applications in breath analysis. In addition, the preliminary but promising results reveal that the CrWO/Ti_3_C_2_ nanocomposite is a potential candidate for a point-of-care breath analysis device for diabetes management and diagnostics.

## Figures and Tables

**Figure 1 biosensors-12-00332-f001:**
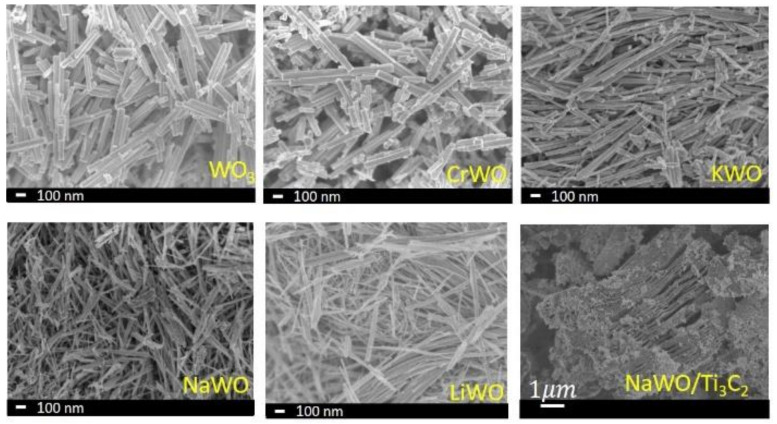
SEM images of WO_3_ and MWO.

**Figure 2 biosensors-12-00332-f002:**
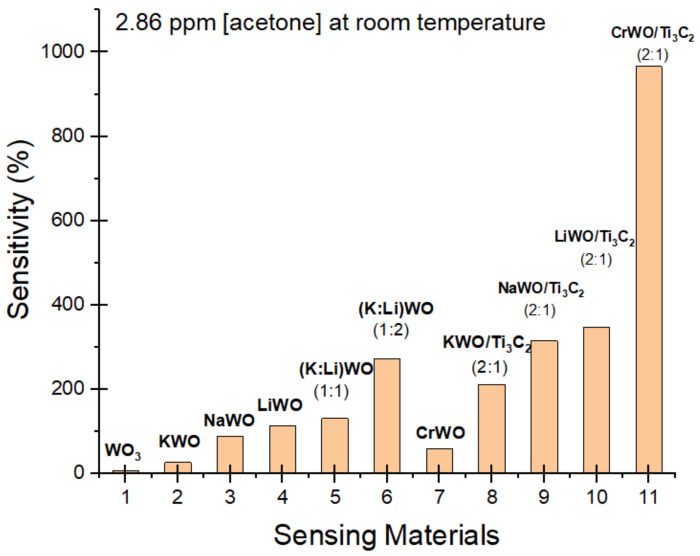
Results of the acetone (2.86 ppm) sensitivity test for various sensors. From left to right, the materials are WO_3_, KWO, NaWO, LiWO, K:Li (1:1), K:Li(1:2), CrWO, KWO/Ti_3_C_2_ (2:1), NaWO/Ti_3_C_2_ (2:1), LiWO/Ti_3_C_2_ (2:1), and CrWO/Ti_3_C_2_ (2:1). All sensors operated at room temperature.

**Figure 3 biosensors-12-00332-f003:**
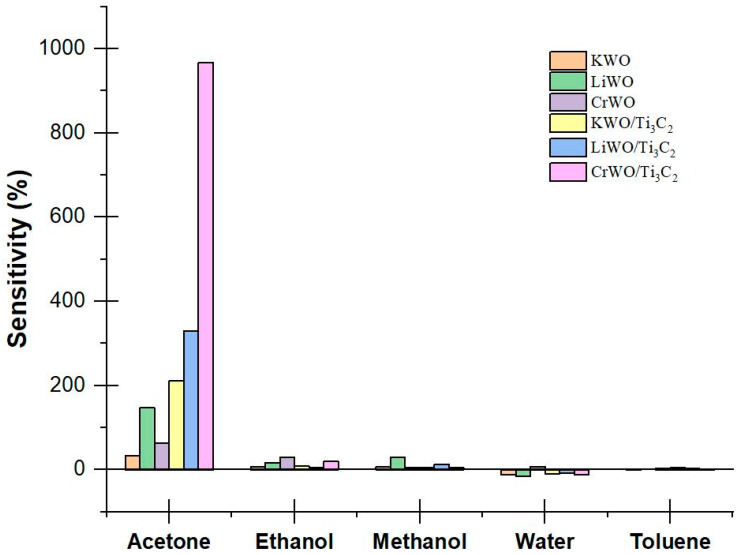
Selectivity data of various materials are shown. The vertical axis represents the sensitivity as defined in previous equations. The chemicals that were tested are shown on the horizontal axis.

**Table 1 biosensors-12-00332-t001:** Testing of various materials in comparison to the data of the materials tested for this paper. The bolded values in the table show results that were the most promising for the required conditions. The CrWO/Ti_3_C_2_ sensor showed high sensitivity at a low operating temperature.

Material	ZnO	Si Doped $WO_{3}$	$SnO_{2}$	W Doped NiO	$Co_{3}O_{4}$	$Cr_{2}WO_{6}$	Mesoporous $WO_{3}$-25 NFs	CrWO/Ti_3_C_2_
Semiconductor Type	n-type	n-type	n-type	p-type	p-type	p-type	n-type	p-type
Lowest Concertation (ppm)	0.1	0.6	5	1	10	0.1	5	**0.1**
Response (R_a_/R_g_)	2.9	5.6	6.7	5.1	1.34	2.3	3.1	**3.44**
Operating Temperature	240	400	260	250	240	300	300	**22**
Ref	[36]	[12]	[37]	[38]	[39]	[40]	[40]	*This Work*

**Table 2 biosensors-12-00332-t002:** Comparison of band gap size to sensitivity percentage to 2.86 ppm acetone for alkali doped WO_3_ at room temperature [44].

Material	$WO_{3}$	LiWO	NaWO	KWO
Bandgap Size (eV)	3.13	2.65	2.65	2.61
Sensitivity (%)	7.95	114.41	89.35	27.38

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
