# Peer review of "Two-Dimensional Ti3C2 MXene-Based Novel Nanocomposites for Breath Sensors for Early Detection of Diabetes Mellitus"

_biosensors, 2022, doi:10.3390/bios12050332_

Round 1

Reviewer 1 Report

The article's authors studied the use of the Ti3C2 MXene-based nanocomposites as the acetone sensor. The manuscript should be revised based on the following comments.

Since the focus of this manuscript is on the use of the materials. The materials should be appropriately characterized. I recommend authors use XPS to characterize the synthesized materials. 

The SEM images of the Ti3C2 MXene-based nanocomposites will reveal the surface property of the material. Morphology is one of the critical factors in designing the sensor.

Sensor fabrication should be added to the experimental section. The sensing area, electrode size, and gap between the interdigitated electrode play a vital role. These parameters need to be updated in the revised manuscript.

The authors have tested sensors at the acetone concentration of 2.867 ppm. How do authors come up with this unique concentration? The sensor's limit of detection and detection range will add value to the reader. 

Table 1: CuO-ZnO is not just a p-type semiconductor; it's p−n junction material.

Figure 3: The author has not mentioned the concentration of each of the VOCs used for testing selectivity. What is the science behind such a strong selectivity?

What is the significance of Table 2? Is there any correlation between the bandgap and the sensitivity?

Author Response

We would like thank you all for the valuable comments provided by the reviewer 1. We have intensively revised the manuscript utilizing their comments, and rewritten paper to address the concerns and comments given by the reviewers. Below are our responses to the comments of the reviewer 1.

Reviewer 2 Report

The present manuscript reported “Two-Dimensional Ti3C2 MXene based Novel Nanocomposites for breath sensors in diabetes mellitus early detection”. This paper is well structured and reports a high sensitive acetone sensor (Figure 1.), but, some parts need to be improved.

  1. please add a new table, and compare the sensitivity of your electrode with other reports.
  2. the title should be to be modified,” Two-Dimensional Ti3C2 MXene based Novel Nanocomposites” actually, CrWO/Ti3C2 is active nanostructures in your electrode.
  3. page 3, 2. Materials and Methods” how did you synthesize “Ti3C2 MXene nanosheets” , please add enough information.
  4. KWO or K2W7O22 ? !!!! check whole the paper, please be careful they are the same. So please modify it

5. please add more information about the role of  “Cr “doped WO3”.

  1. page3, “Our preliminary results revealed the 1D/2D MWO/Ti3C2 nanocomposites”, please add the related references.
  2. the main problem is abstract, please improve it, low information! no data! No your findings!
  3. please add an image from your sensor, and analytical system.
  4. to improve your work please these references//////Yoon, Jinho, Minkyu Shin, Joungpyo Lim, Ji-Young Lee, and Jeong-Woo Choi. "Recent advances in MXene nanocomposite-based biosensors." Biosensors 10, no. 11 (2020): 185.//////Tomić, Milena, Milena Šetka, OndÅ™ej Chmela, Isabel Gràcia, Eduard Figueras, Carles Cané, and Stella Vallejos. "Cerium oxide-tungsten oxide core-shell nanowire-based microsensors sensitive to acetone." Biosensors 8, no. 4 (2018): 116./////Hatamie, A., Angizi, S., Kumar, S., Pandey, C.M., Simchi, A., Willander, M. and Malhotra, B.D., 2020. Textile based chemical and physical sensors for healthcare monitoring. Journal of the Electrochemical Society, 167(3), p.037546.//////Sadiq, Mahek, Lizhi Pang, Michael Johnson, Venkatachalem Sathish, Qifeng Zhang, and Danling Wang. "2D Nanomaterial, Ti3C2 MXene-based sensor to guide lung cancer therapy and management." Biosensors 11, no. 2 (2021): 40.

Author Response

We would like thank you all for the valuable comments provided by the reviewer 2. We have intensively revised the manuscript utilizing their comments, and rewritten paper to address the concerns and comments given by the reviewers. Below are our responses to the comments of the reviewer 2.

Round 2

Reviewer 1 Report

The authors have addressed all the comments and concerns related to this paper. As a result, the paper has been significantly improved. Thus, this can be accepted in its present form.  

Reviewer 2 Report

it is really improved. please accept it.